# Compositional Features and Nutritional Value of Pig Brain: Potential and Challenges as a Sustainable Source of Nutrients

**DOI:** 10.3390/foods10122943

**Published:** 2021-11-30

**Authors:** Jaruwan Chanted, Worawan Panpipat, Atikorn Panya, Natthaporn Phonsatta, Ling-Zhi Cheong, Manat Chaijan

**Affiliations:** 1Food Technology and Innovation Research Center of Excellence, School of Agricultural Technology and Food Industry, Walailak University, Nakhon Si Thammarat 80160, Thailand; Jaruwanchanted@gmail.com (J.C.); pworawan@wu.ac.th (W.P.); 2Food Biotechnology Research Team, Functional Ingredients and Food Innovation Research Group, National Center for Genetic Engineering and Biotechnology (BIOTEC), National Science and Technology Development Agency, Bangkok 12120, Thailand; atikorn.pan@biotec.or.th (A.P.); natthaporn.pho@biotec.or.th (N.P.); 3Zhejiang-Malaysia Joint Research Laboratory for Agricultural Product Processing and Nutrition, College of Food and Pharmaceutical Science, Ningbo University, Ningbo 315211, China; cheonglingzhi@nbu.edu.cn

**Keywords:** pork, fatty acid, amino acid, mineral, meat, by-product, sustainability

## Abstract

The goal of this study was to establish the nutritional value and compositional properties of the brains of crossbred pigs (Landrace–Large white–Duroc (LLD)), in order to realize the zero-waste concept and increase the use of by-products in the sustainable meat industry. Fat (9.25% fresh weight (fw)) and protein (7.25% fw) were the principal dry matters of pig brain, followed by carbohydrate and ash. Phospholipid and cholesterol had a 3:1 ratio. Pig brain had a red tone (*L** = 63.88, *a** = 5.60, and *b** = 15.43) and a high iron content (66 mg/kg) due to a total heme protein concentration of 1.31 g/100 g fw. The most prevalent macro-element was phosphorus (14 g/kg), followed by potassium, sodium, calcium, and magnesium. Zinc, copper, and manganese were among the other trace elements discovered. The most prevalent nitrogenous constituents were alkali-soluble protein, followed by water-soluble protein, stromal protein, salt-soluble protein, and non-protein nitrogen. Essential amino acids were abundant in pig brain (44% of total amino acids), particularly leucine (28.57 mg/g protein), threonine, valine, and lysine. The total lipid, neutral, and polar lipid fractions of the pig brain had different fatty acid compositions. The largest amount was observed in saturated fatty acids (SFA), followed by monounsaturated fatty acids (MUFA) and polyunsaturated fatty acids (PUFA). Stearic acid and palmitic acid were the most common SFA. Oleic acid was the most prevalent MUFA, while docosahexaenoic acid was the most common PUFA. Thus, the pig brain can be used in food formulations as a source of nutrients.

## 1. Introduction

Pork consumption is currently on the rise among the world’s population. China, the European Union, the United States, Brazil, and Russia are the world’s top five pork producers [1]. Thailand’s pig production was estimated to be over 20.5 million heads in 2020, with pork consumption estimated to be around 1.3 million tons [1]. By-products, such as blood, bone, bristle, fat trimmings, viscera, and brain, are created as a result of increased pig consumption [2,3,4]. By-products constitute between 60% and 70% of the butchered carcass, with roughly 40% being edible and 20% being inedible [4]. Some of these by-products are widely used in many countries around the world in a variety of traditional dishes [3] and can have value effectively added using additional processes, such as thermal, chemical, centrifugation, washing, and combined processes to produce lard, flavor concentrate, plasma, red blood cell, gelatin, protein hydrolysates, and other products [4,5,6,7]. However, a range of factors, such as religion, culture, income, and personal taste, have an impact on the utilization of meat by-products. Depending on the country and local traditions, various meat by-products can be considered edible in certain locations but inedible in others. In reality, high-nutrient by-products such as liver, heart, blood, lung, spleen, kidney, brains, and tripe are used in the cuisines of some countries around the world [3]. Naturally, the nutritional makeup of each by-product is dependent on the animal type from which it is derived [3]. The basic composition and nutritional worth of these by-products should be assessed first, in order to find approaches to boost their value.

Although pig brain is an important by-product of slaughtering and pork processing, it has not yet been extensively used, particularly for human consumption. There is also no academic knowledge on how to increase the value of pig brain. Only traditional cookery, such as soup, gravy, stew, curry, and fried food, has been recognized as a primary manner of using the pig brain. When using pig brain as food, not only should the issue of safety, particularly of prion diseases, be considered, but the chemical compositions and nutritive value must also be assessed in order to provide nutritional information and pave the way for improved exploitation of the pig brain [8]. As a result, the goal of this study was to determine the nutritional value and compositional features of pig brain. The findings may be valuable in boosting pig brain intake and subsequent exploitation in the sustainable meat sector in order to achieve zero waste.

## 2. Materials and Methods

### 2.1. Chemicals

All chemicals and reagents used in this study, e.g., trichloroacetic acid (TCA), sodium dodecyl sulfate (SDS), potassium hydroxide (KOH), and potassium chloride (KCl), were acquired from Sigma-Aldrich (St. Louis, MO, USA).

### 2.2. Collection and Preparation of Pig Brains

Ten brains of crossbred pigs (Landrace–Large white–Duroc, LLD) at 4 months of age were collected from the Shaw Processing Food Co., Ltd. in Nakhon Si Thammarat, Thailand. The brains came from healthy pigs, approved by Thailand’s Bureau of Livestock Standards and Certification. Within 1 h, the obtained samples were delivered to Walailak University’s Food Technology and Innovation Laboratory in ice with a sample-to-ice ratio of 1:2 (*w/w*). The brains were then rinsed in cold water (4 °C), drained, and chopped with a Talsa Bowl Cutter K15e (The Food Machinery Co., Ltd., Kent, UK) to create a homogeneous composite sample. After vacuum packing (DZQ-400, Afapa Vacuum Equipment Co., Ltd., Shanghai, China), ground samples were kept at −80 °C for no more than 1 month before being utilized experimentally.

### 2.3. Proximate Composition

Moisture (A.O.A.C method number 950.46), crude protein (A.O.A.C method number 928.08, Kjeldahl factor of 6.25), fat (A.O.A.C method number 963.15), ash (A.O.A.C method number 920.153), and carbohydrate (calculated by the difference) were all investigated in the proximate composition of pig brain [9]. The results were expressed in grams per 100 g of fresh weight (fw).

### 2.4. Determination of Total Phospholipid and Total Cholesterol Contents

Bligh and Dyer’s method [10] was used to extract lipid from the pig brain. Samples (25 g) were homogenized with 200 mL of a mixture of chloroform, methanol, and distilled water (1:2:1, *v/v/v*) at 9500 rpm for 2 min at 4 °C using an IKA Labortechnik homogenizer (Selangor, Malaysia). Then, 50 mL chloroform was added to the homogenate, and the mixtures were homogenized at the same speed for 1 min. After that, 25 mL distilled water was added, and the mixtures were homogenized at the same speed for 30 s. The homogenate was centrifuged at 3000× *g* for 15 min at 4 °C using an RC-5B plus centrifuge (Sorvall, Norwalk, CT, USA) and then transferred to a separating flask. The chloroform phase was drained into a 125 mL Erlenmeyer flask containing around 2–5 g of sodium sulfate, agitated well, and filtered through Whatman No. 4 filter paper (GE Healthcare Bio-Sciences Corp., Pittsburgh, PA, USA) into a round-bottom flask. A rotary evaporator (Model N-100, Eyela Ltd., Tokyo, Japan) was used to evaporate the solvent at 40 °C.

The total phospholipid content of the extracted oil was determined using a modified Stewart method [11]. Oil samples (20 μL) were dissolved in chloroform to obtain a final volume of 2 mL. Then, 1 mL of thiocyanate reagent (a mixture of 0.10 M ferric chloride hexahydrate and 0.40 M ammonium thiocyanate) was added. The lower layer was removed after 1 min of vigorous mixing and the absorbance at 488 nm was determined. Phosphatidylcholine (0–50 ppm) was used to create a standard curve. The total phospholipid content was measured in g/100 g fw.

The cholesterol content of the oil samples was determined using a modified version of Beyer and Jensen’s method [12]. The oil sample (0.1–0.2 g) was saponified using 2% alcoholic KOH for 10 min. The unsaponified fraction was extracted with 2 × 10 mL hexane. The extracts were rinsed with 5 mL distilled water and dried at 45 °C in a W350 Memmert temperature-controlled water bath (Schwabach, Germany). The dried extract was resuspended in 3 mL of glacial acetic acid, and 2 mL coloring reagent was added. To prepare the coloring reagent, the stock reagent of 10% (*w/v*) FeCl_3_·6H_2_O in glacial acetic acid was made and then 1 mL of the stock reagent was diluted with 100 mL of concentrated H_2_SO_4_. The absorbance of the reaction mixture was read at 565 nm against a glacial acetic acid blank using a UV–Vis spectrophotometer (UV-1900, Shimadzu, Kyoto, Japan). A standard curve was prepared using cholesterol in glacial acetic acid at 0 to 120 mg/L. The total cholesterol content was measured in g/100 g fw.

### 2.5. Total Heme Protein Content and Color Measurement

Using Chaijan and Undeland’s method [13], the total heme protein content was assessed and stated in g of hemoglobin per 100 g of sample. A sample and 3 volumes of 0.1 M phosphate buffer, pH 7 containing 5% SDS (*w/v*) was homogenized at 13,500 rpm for 20 s. The homogenate was heated in a water bath (85 °C) for 1 h and cooled under running tap water for 10 min. The solution was then centrifuged (5000× *g*/15 min/25 °C). The absorbance of the supernatant was read at 535 nm using a UV–Vis spectrophotometer with phosphate buffer as a blank. A standard curve of bovine hemoglobin (0–20 µM) was used.

A portable Hunterlab ColorFlex^®^ EZ device (Hunter Assoc. Laboratory; Reston, VA, USA) was used to collect colorimetric data of the pig brain in triplicate. A white and black standard were used to calibrate the device. The measurement modes tristimulus *L** (lightness), *a** (redness/greenness), and *b** (yellowness/blueness) were chosen. According to Chen et al. [14], the redness index (*a**/*b**) was computed.

### 2.6. Mineral Composition

The mineral composition including phosphorus (P), potassium (K), sodium (Na), calcium (Ca), magnesium (Mg), iron (Fe), zinc (Zn), copper (Cu), manganese (Mn), and chromium (Cr) was determined [9]. Samples (4 g) were mixed with 4 mL of strong nitric acid and vigorously shaken for 5 min. The mixtures were heated on a hot plate until digestion was completed. The digested samples were transferred to a volumetric flask and filled to a capacity of 10 mL with deionized water. An inductively coupled plasma optical emission spectrophotometer was used to analyze the solution (PerkinElmer, Model 4300DV, Norwalk, CT, USA). The flow rates of argon to plasma, auxiliary, and nebulizer were kept at 15, 0.2, and 0.8 L/min, respectively. The sample’s flow rate was set at 1.5 mL/min. The mineral content was determined and expressed in mg/kg.

### 2.7. Protein Fractionation

The protein composition of pig brain was fractionated according to Hashimoto et al. [15]. The non-protein nitrogenous (NPN) compound fraction, water-soluble protein fraction, salt-soluble protein fraction, alkali-soluble protein fraction, and stromal protein fraction were separated from the pig brain proteins due to their varied solubilities. Briefly, the ground sample (20 g) were homogenized in 200 mL of phosphate buffer (15.6 mM Na_2_HPO_4_, 3.5 mM KH_2_PO_4_), pH 7.5 with an IKA Labortechnik homogenizer. The homogenate was centrifuged (5000× *g*/15 min/4 °C) using an RC-5B plus centrifuge. The residue was homogenized and centrifuged again after being mixed with 200 mL of the same buffer. These two supernatants were combined, and TCA was added to obtain a final concentration of 5% (*w/v*). The resulting precipitate was collected by filtration and referred to as the water-soluble protein fraction. The filtrate was used as the NPN fraction. For the above residue, 10 vol of phosphate buffer (15.6 mM Na_2_HPO_4_, 3.5 mM KH_2_PO_4_) containing 0.45 mM KCl, pH 7.5 was added. The mixture was homogenized and centrifuged (5000× *g*/15 min/4 °C). The procedure was carried out twice more. Both supernatants were combined and used as the salt-soluble protein fraction. The precipitate obtained was added with 5 vol of 0.1 M NaOH and stirred for 10 h at 4 °C. The mixtures were then centrifuged (5000× *g*/15 min/4 °C). The supernatant was given as the alkali soluble protein fraction. The final precipitate was used as the stromal protein fraction. The nitrogen distribution was estimated after each fraction by the Kjeldahl method [9].

### 2.8. Amino Acid Profile and Fourier Transform Infrared (FTIR) Spectroscopy

According to Chinarak et al. [16], the amino acid profile of the pig brain was measured. In a 10 mL crimp seal glass vial, freeze-dried samples were combined with 5 mL of hydrolysis solution (6 M HCl, 5% thioglycolic acid, and 1% phenol) and tightly sealed. One 1 mL of the sample was centrifuged at 10,000× *g* for 10 min) after being hydrolyzed at 110 °C for 18 h. The supernatant (100 μL) was then neutralized with 1 M sodium carbonate. An aliquot of 25 μL was transferred to a 2 mL GC glass vial, which was then filled with 50 μL of 200 nM norleucine as an internal standard. After drying for around 1–2 h at 60 °C, 50 μL of dichloromethane was added and dried for another 30 min to eliminate any remaining water. Then, a derivatizing agent, *N*-tert-Butyldimethylsilyl-*N*-methyltrifluoroacetamide, with 1% *tert*-Butyldimethylchlorosilane (50 µL) and acetonitrile (50 µL), were mixed with the samples and subjected to incubate in a hot-air oven (100 °C/4 h). The sample (2 μL) was analyzed using a Shimadzu GCMS-TQ8050 NX (Kyoto, Japan) after cooling to room temperature.

A horizontal attenuated total reflectance (ATR) trough plate crystal cell (45° ZnSe; 80 mm long, 10 mm wide and 4 mm thick) (Pike Technology, Inc., Madison, WI, USA), equipped with a Bruker Model Vector 33 FTIR spectrometer (Bruker Co., Ettlingen, Germany), was used to perform FTIR analysis on the freeze-dried pig brain and pig brain lipid. In the mid-infrared region (500–4000 cm^−1^), 16 scans, at a resolution of 4 cm^−1^, were used to capture FTIR spectra at room temperature (26–29 °C). A reference air spectrum was acquired as a background. Analysis of spectral data was carried out using the OPUS 3.0 data collection software program [17].

### 2.9. Fatty Acid Profile of Total Lipid, Neutral Lipid Fraction, and Polar Lipid Fraction

According to Estefanell et al. [18], the neutral and polar fractions of total lipids were fractionated by adsorption chromatography on silica cartridges (Sep-pak; Waters S.A., MA, USA) using 30 mL chloroform and 20 mL chloroform/methanol (49:1, *v/v*) as neutral lipid solvents, followed by a 30 mL methanol rinse to yield the polar fraction. Fatty acid methyl esters (FAME) in the samples (total, neutral, and polar lipids) were determined using a gas chromatography/quadrupole time of flight (GC/Q-TOF) mass spectrometer (GC 7890B/MSD 7250, Agilent technologies, USA) coupled to the PAL auto sampler system (CTC Analytics AG, Switzerland). The MassHunter software was used to collect MS data (Version 10.0, Agilent Technologies, Santa Clara, CA, USA). The complete method and optimal condition can be obtained from the report of Chinarak et al. [16].

### 2.10. Statistical Analyses

All analyses were run in triplicate. An analysis of variance was performed on the data. Duncan’s multiple-range test was used to compare the means. SPSS 17.0 for Windows (SPSS Inc., Chicago, IL, USA) was used to conduct the statistical analysis.

## 3. Results and Discussion

### 3.1. Proximate Composition

The approximate analysis of the pig brain is shown in Table 1. Moisture was the most abundant composition in the pig brain sample, accounting for 79.96% (fw), alike to other pork by-products [19]. Among the dry matter, fat was the most abundant composition (9.25% fw), followed by protein (7.25% fw), carbohydrate (2.21% fw), and ash (1.33% fw), respectively. Lipids and their intermediates are important parts of the brain’s structure and function. Behind adipose tissue, the brain has the second largest lipid content, with lipids accounting for half of the brain’s dry weight [20]. Krafft et al. [21] reported that brain tissue is composed of around 70–83% water, 7.5–8.5% protein (fw), and 5–15% lipid (fw). Brain tissue contained high content of lipid, which was mainly divided into three major groups, including neutral lipids, phospholipids, and sphingolipids [21]. Pig brain had considerably higher total lipid content than fresh pork loin (1.27–3.41% fw) [22] and pork by-products, such as heart, liver, lung, stomach small intestine, large intestine, spleen, uterus, and pancreas (0.28–7.18% fw) [19]. The protein content of the pig brain, on the other hand, was lower than that of fresh pork (19.80%, fw) [23] and other pork by-products, particularly viscera portions (8.45–22.05% fw), which have been reported in the literature [19]. The results revealed that the pig brain could be a rich source of nutrients, especially lipid and protein. The total protein and lipid content of the original material are both important factors in deciding how to use and recover by-products [24,25].

### 3.2. Total Phospholipids and Cholesterol Contents

Phospholipids are functional and structural components of cell membranes that are easily absorbed by the body and therefore perform their functions. Phospholipids are a dietary supply of lipoproteins, which play a significant role in lipid metabolism. Dietary phospholipid has been shown, in mammalian studies, to limit lipid deposition in the liver by blocking lipid absorption and oxidation [26,27]. Phospholipid is one of the most abundant lipids in brain tissue [21]. From this study, the phospholipid content of the oil extracted from pig brain was 0.86 g/100 g fw. The phospholipid content of crude lipid from pig brain was similar to that of alternative foods such as sago palm weewil (*Rhynchophorus ferregineus*) larvae (2.6–9.3 g/100 g lipid) [28], but it was lower than that of popular ingredients, such as fish oil (13.7–32.9 g/100 g lipid) [29] and krill oil (32.5 g/100 g lipid) [30]. Phospholipid is a natural surfactant that may be used to prepare emulsions and has high emulsifying characteristics in general [31]. Phospholipids can also act as an antioxidant [32]. Phospholipids can form associations with cholesterol in cell membranes [33]. Cholesterol is also a significant constituent of brain cell membranes [34]. The typical quantity of cholesterol found in an animal is around 2.1 g/kg of body weight [35]. Table 1 shows that the total cholesterol concentration of the pig brain was 0.30 g/100 g fw, resulting in a phospholipid to cholesterol ratio of around 3:1. Pig brain lipids had a lower cholesterol level (3.27 g/100 g lipid) than certain typical foods, such as cooked duck egg (6.4 g/100 g lipid) and raw squid (16.9 g/100 g lipid) [28]. Cholesterol is a lipid produced by the liver of animals and found in all animal-based foodstuffs, including eggs, red meat, and fish [36]. Cholesterol, which is one of the main structural components of cell membranes and is turned into hormones, is essential to good health at a normal level [37]. Hypercholesterolemia, cardiovascular disease, and coronary heart disease are all linked to a high-cholesterol diet [37]. The adult population should consume no more than 300 mg of cholesterol each day [28]. Although pig brain lipids are high in phospholipids, the cholesterol content should be considered when consuming pig brain. Separating cholesterol from pig brain oil could be an approach for increasing the oil’s use as a functional ingredient.

### 3.3. Total Heme Protein and Color

The pig brain has a total heme protein concentration of 1.31 g/100 g fw. This was attributed to the presence of a blood vessel in the brain, which is evident in Table 1. Blood vessels, particularly capillaries, form a dynamic and intricate architecture that transports oxygen and nutrients to the brain [38]. As food, the presence of heme protein in form of blood may influence the storage stability, specifically, discoloration, rancidity development, and microbial growth [13,39,40]. Positive *a** (5.60) and *b** (15.43) values with a redness index of 0.36 indicated that the presence of heme protein caused the pig brain to be reddish brown in color (Table 2). The high moisture content in the pig brain (Table 1) was represented by a high *L** (63.88) value (Table 2). The elevated redness levels could be explained by the presence of heme proteins. The amount of moisture, membrane protein, connective tissue, lipid, and heme proteins in pig brain may affect its color. Moisture, membrane protein, and connective tissue can cause lightness; lipid can give yellowness; heme proteins (e.g., hemoglobin) can cause redness; and hemoglobin oxidation/denaturation can cause a yellow-brownish color.

### 3.4. Mineral Composition

As shown in Table 3, pig brain is a source of macro and trace elements. Minerals’ value as dietary ingredients is based on more than just their nutritional and physiological functions. By stimulating or suppressing enzyme-catalyzed and other reactions, minerals contribute to food flavor, color, and texture [41]. P (14.0 g/kg) was found to be more abundant than the other macro-elements, followed by K (9.6 g/kg), Na (5.6 g/kg), Ca (2.2 g/kg), and Mg (0.7 g/kg). Mineral concentrations in pig brain were higher than those seen in other pork by-products [19]. Free element, salt, phosphoproteins, and phospholipids are all examples of P. Pig brain phosphoproteins with calcium-binding capacity have been isolated [42]. K is required for the maintenance of cell membrane potentials, and a lack of it is linked to hypertension and an increased risk of cardiovascular disease [43]. The major extracellular osmolyte, serum Na, is the most essential predictor of serum osmolality, including the central nervous system of the brain [44]. The pig brain had a lower Ca concentration than the major Ca source in human diet—cows’ milk [43]—but it could be another source of Ca. Mg is important for nerve transmission and muscle conduction from a neurological standpoint. It also protects neurons against excessive excitement, which can cause neuronal cell death, and has been linked to a variety of neurological illnesses [45].

Fe had the highest concentration of trace elements observed, followed by Zn, Cu, and Mn. Cr was not found in the pig brain (Table 3). Fe was found to be the most abundant trace element in pig visceral by-products, according to Seong et al. [19]. These trace minerals are necessary for human health, and a lack of them can cause nutritional deficiency symptoms [46]. Trace elements are molecular micronutrients that are needed in minute amounts but are crucial for the viability of many physiological functions in living tissues [41]. Pig brain Fe, Zn, and Cu levels were higher than those seen in other pork by-products [19]. Fe is the most abundant essential trace element in the human body, and it is a necessary micronutrient for brain development. Normal brain growth, myelination, and neurotransmission all require Fe [47]. The overall amount of Fe in the body is around 3–5 g, with the majority of it in the blood and the rest in the liver, bone marrow, and muscles [41]. Fe is one of the key nutrients for blood’s optimal function; anemia, especially in pregnant women and children, is caused by a lack of Fe [48]. The pigment level in the pig brain may be caused by high Fe concentrations. The prostate, portions of the eye, the muscle, brain, bones, kidneys, and liver all store zinc [49]. After Fe, Zn is the second most prevalent transition metal in organisms [41]. Cu is found in practically all physiological tissues and is mostly stored in the liver, as well as the brain, heart, kidneys, and muscles [50]. Mn is a mineral that aids in the growth of the body, metabolism, and enzymatic defense mechanisms [51].

### 3.5. Distribution of Nitrogenous Constituents and Amino Acid Profile

Table 4 shows the distribution of nitrogenous constituents in pig brain. The ratio of NPN to protein N was roughly 1:28. Alkali-soluble protein accounted for 47.79% of the protein N, followed by water-soluble protein (21.60%), stroma (21.10%), and salt-soluble protein (9.49%). The presence of alkali-soluble protein in the membrane has been identified [52]. Proteins undergo modifications in alkaline conditions, which drive them apart by repulsion, allowing for associations with water and therefore solubilization [53,54]. In the cytosolic fluid of the brain, water-soluble and salt-soluble proteins may be found [55]. Complexins, for example, are a cytosolic protein family [56]. Collagen has been discovered in the brain as one of the principal stroma [57], particularly in the capillary walls [38,58].

The pig brain was found to be a good supply of essential amino acids (EAA), based on the findings (Table 4). The nutritional quality of food proteins is governed by the composition, quantity, and availability of EAA [59]. The total EAA of pig brain protein was around 44% of total amino acid, which was higher than the joint WHO/FAO/UNU expert consultation’s recommendation of around 29% [60]. The most abundant EAA in pig brain was leucine (28.57 mg/g), followed by threonine, valine, lysine, isoleucine, phenylalanine, histidine, and methionine. Valine, phenylalanine, lysine, histidine, leucine, and isoleucine were the most common EAA in pig liver and heart [19]. EAA profiles of pig brain protein were similar to those of animal proteins such as milk, egg, fish, and meat [59,61].

The most prevalent non-essential amino acid (NEAA) was glutamic acid (44.11 mg/g), followed by aspartic acid, serine, glycine, alanine, proline, threonine, and cysteine. Glutamic acid, aspartic acid, glycine, and alanine are classified as umami-taste active amino acids (UAA), which play a role in the umami flavor [59,62,63]. The UAA level of pig brain was around 36% of total amino acid (Table 4). The UAA concentration of pig brain was similar to that of farm-raised sturgeon caviar (37.44–38.04%) and seaweeds (37.59–42.50%) [59], but it was higher than that of wild edible mushrooms (7–22%) [64]. As a result, the umami flavor of the pig brain could be detectable. Umami-relevant substances in pork *longissimus* and *biceps femoris* muscles include glutamic acid, total free amino acid, inosine monophosphate (IMP), and soluble oligopeptides [65]. Functional amino acid (FAA) are amino acids found in living beings that have a role in and control critical metabolic pathways [66]. FAA can be EAA or NEAA and is composed of arginine, aspartic acid, cysteine, glutamic acid, glycine, leucine, methionine, proline, tryptophan, and tyrosine [66]. FAA aids in the treatment of some metabolic abnormalities and the modulation of the immune system [59,66]. The presence of FAA is significant due to their critical biological roles. In pig brain, FAA was detected in significant amounts (67.41%) (Table 4).

The amount and arrangement of hydrophobic and hydrophilic amino acids, particularly the interfacial characteristics, are critical to protein techno-functionality [63]. The hydrophobic amino acids made up 33.85% of the total amino acids in the pig brain. The ability of a protein to operate as an emulsifying agent should be linked to its hydrophobicity. Furthermore, the solubility profile of pig brain protein may be influenced by its amino acid composition. Protein solubility has been shown to be influenced by hydrophilic amino acids such as glutamic acid and aspartic acid [67]. As a result, pig brain can be exploited as a nutritious source of protein with potential technical applications.

### 3.6. FTIR Spectra

Figure 1 shows the FTIR spectra of whole pig brain and its lipid. The whole pig brain and its lipids have “fingerprint” spectra in the wavenumber range of 2700 to 3500 cm^−1^, which are originated from stretch vibrations of CH, NH, and OH groups and related to their chemical compositions [21]. In the range of 400 to 1800 cm^−1^, the spectra of whole pig brain and lipid fraction differed, indicating the presence of distinct chemical bonds and functional groups in the samples. In the whole pig brain, the various bands revealed the presence of water, protein, lipid, and carbohydrate. Some peaks may, however, be overlapping due to the combination of various compositions. According to Guillén and Cabo [68] and Lerma-García et al. [69], the presence of the vibration OH stretching of water allows the first band (3600–3200 cm^−1^) to function. The peaks at 2850–2950 cm^−1^ correspond to the CH stretching bonds of methyl and methylene. The peaks at 1738 cm^−1^ are due to the presence of triglyceride functional groups (C=O stretching). The amide I peak at 1655 cm^−1^ (C=O stretching/hydrogen bonding paired with CN stretch and CCN deformation), the amide II peak at 1547 cm^−1^ (NH ending mixed with CN stretching), and the amide III peak at 1452 cm^−1^ (CN stretching and NH deformation) are all characteristic of proteins [70,71]. The amide A and amide B peaks were also identified at 3282 and 2922 cm^−1^, respectively [63]. Finally, carbohydrates are responsible for the peaks in the 900–1200 cm^−1^ range [72], with the most conspicuous saccharide band about 1370 cm^−1^ [21].

In the spectra of lipids, due to the purity of the lipid and their lipid composition (e.g., triglyceride, phospholipid, cholesterol, sphingomyelin, and cerebrosides), the peaks in the range 400 to 1800 cm^−1^ were different from those of the original whole pig brain. Attributed to the existence of ester groups, Krafft et al. [21] found that the neutral lipid triacylglyceride showed a peak at 1729 cm^−1^. The existence of phospholipids could explain the presence of a high peak at 835 cm^−1^. It has been reported that phosphatidic acid, the parent component of phospholipids, has an 860 cm^−1^ band [21]. Several phospholipid varieties, including phosphatidylcholine (PC), phosphatidylethanolamine (PE), phosphatidylserine (PS), and phosphatidylinositol (PI), were discovered in the brain, each with a different spectrum, due to differences in functional head groups or fatty acid residues [21]. According to Krafft et al. [21], the cholesterol spectrum has multiple sharp bands ranging from 400 to 1200 cm^−1^, with the most intense ones at 429, 548, 608, and 702 cm^−1^. Furthermore, the peak at 1440 cm^−1^ is attributed to cholesterol CH distortion. In the spectra of sphingomyelin, choline bands are positioned at 718 and 875 cm^−1^, the same as in PC, because sphingomyelin has a PC residue linked to the ceramide backbone [21].

### 3.7. Fatty Acid Profiles

Lipids are important parts of the brain’s structure and function. Essential fatty acids must be delivered into the brain from the circulation, despite the fact that some fatty acids can be produced de novo [73]. Unlike adipose tissue, which stores fatty acids predominantly as triglycerides, the brain is thought to create phospholipids for cell membranes primarily from acylated lipids [20]. The components of brain lipids can be separated into two categories: neutral lipid and polar lipid [21]. As a result, the fatty acid composition of total lipid was compared with that of neutral and polar lipid fractions. It was discovered that there is a difference in fatty acid profiles in pig brain between the total lipid and its fractions (Table 5). In general, the brain’s fatty acid makeup is distinctive, with a high concentration of long-chain polyunsaturated fatty acids (PUFA), such as eicosapentaenoic acid and docosahexaenoic acid (DHA) [73]. Mas et al. [74] and Seong et al. [19] reported on the fatty acid content of several pork by-products and found that there was a considerable variance in the fatty acid composition.

From the results, saturated fatty acid (SFA) was found to have the highest content, followed by monounsaturated fatty acid (MUFA) and polyunsaturated fatty acid (PUFA) (*p* < 0.05). The most prevalent SFAs were stearic acid (C18:0) and palmitic acid (C16:0). The results were in agreement with Seong et al. [19], who reported that palmitic acid and stearic acid were the main SFAs found in pork by-products. The most prevalent MUFA was oleic acid (C18:1), while the most common PUFA was DHA (*p* < 0.05). DHA is a kind of *n*-3 PUFA that provides a variety of health effects. DHA was only discovered in the heart, liver, and spleen in a prior investigation by Seong et al. [19], with the liver having the greatest quantity (3.47%). Therefore, pig brain had a higher DHA content than other pork by-products.

MUFAs and PUFAs have long been thought to be beneficial fats, especially when used to replace saturated fats in the diet. This helps to keep blood cholesterol levels in the normal range and has been connected to a lower risk of coronary heart disease [75]. Total *n*-3 PUFA to total *n*-6 PUFA ratios in polar lipid fraction, neutral lipid fraction, and total lipid were 4.72, 5.16, and 6.12, respectively. The pig brain contains a considerable number of important *n*-3 fatty acids, as seen by the relatively high *n*-3/*n*-6 PUFA ratio. Because of their high bioavailability, *n*-3 fatty acid-containing phospholipids have a considerable preventive effect against cardiovascular disease, antioxidant activity, memory improvement, and other possible health advantages [76].

## 4. Conclusions

Important nutrients, such as phospholipids, proteins, essential amino acids (leucine, threonine, valine, lysine, phenylalanine, isoleucine, histidine, and methionine), DHA, and minerals (e.g., P, K, Ca, Mg, Fe, Zn, and Cu), were found in the pig brain. The pig brain in this experiment also included a significant amount of UAA, which can be employed to improve the sensory qualities of pig brain-based products. As a result, the pig brain could be used as a source of nutrients in food preparations. On the other hand, the pig brain exhibited high quantities of SFA and cholesterol, which should be taken into account. To properly investigate the usage of pig brain for human nutrition, such components may need to be removed. To achieve the zero-waste goal, the cholesterol and SFA fractions, on the other hand, may be used in cosmetics in the future. Using the optimal refining approach, the nutritious protein with techno-functionality from pig brain may be isolated, along with the lipid separation for food sustainability.

## Figures and Tables

**Figure 1 foods-10-02943-f001:**
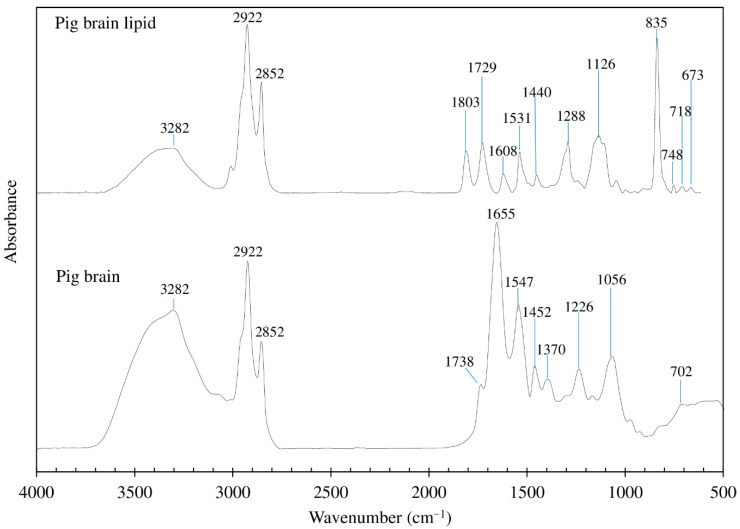
Fourier transform infrared (FTIR) spectra of the freeze-dried pig brain and pig brain lipid.

**Table 1 foods-10-02943-t001:** Chemical composition of pig brain.

Compositions (g/100 g Fresh Weight)	Values
Moisture	79.96 ± 0.19
Fat	9.25 ± 0.32
Protein	7.25 ± 0.96
Carbohydrate	2.21 ± 0.42
Ash	1.33 ± 0.16
Total phospholipid	0.86 ± 0.00
Total cholesterol	0.30 ± 0.01
Total heme protein	1.31 ± 0.03

Values are given as mean ± standard deviation from triplicate determinations.

**Table 2 foods-10-02943-t002:** Color attributes and appearances of pig brain.

Attributes	Values
Color	
*L**	63.88 ± 0.25
*a**	5.60 ± 0.40
*b**	15.43 ± 0.63
Redness index	0.36 ± 6.95
Appearance	
Whole	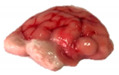
Ground	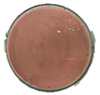

Values are given as mean ± standard deviation from triplicate determinations. *L** = lightness, *a** = redness/greenness, *b** = yellowness/blueness.

**Table 3 foods-10-02943-t003:** Mineral profile of pig brain.

Mineral	Contents (mg/kg)
Macro-element	
Phosphorous (P)	14,021.76 ± 97.20
Potassium (K)	9635.85 ± 132.18
Sodium (Na)	5624.04 ± 23.28
Calcium (Ca)	2196.99 ± 53.57
Magnesium (Mg)	699.39 ± 3.91
Trace-element	
Iron (Fe)	65.79 ± 0.66
Zinc (Zn)	37.90 ± 0.51
Copper (Cu)	12.42 ± 0.13
Manganese (Mn)	0.99 ± 0.01
Chromium (Cr)	nd

nd: not detected. Values are given as mean ± standard deviation from triplicate determinations.

**Table 4 foods-10-02943-t004:** Distribution of nitrogenous constituents and amino acid compositions of pig brain.

Compositions	Values
Nitrogenous constituents (mg N/g fresh weight)	
Protein nitrogen	
Alkali-soluble protein	15.15 ± 0.40 (47.79%) *
Water-soluble protein	6.85 ± 0.00 (21.60%)
Stromal protein	6.69 ± 0.00 (21.10%)
Salt-soluble protein	3.01 ± 0.40 (9.49%)
Non-protein nitrogen (NPN)	1.15 ± 0.40
Amino acid compositions (mg/g protein)	
Essential amino acid (EAA)	
Leucine	28.57 ± 0.46
Threonine	25.50 ± 0.19
Valine	17.76 ± 0.17
Lysine	16.06 ± 0.02
Phenylalanine	12.89 ± 0.07
Isoleucine	12.57 ± 0.11
Histidine	8.70 ± 0.11
Methionine	7.56 ± 0.10
Non-essential amino acid (NEAA)	
Glutamic acid/glutamine	44.11 ± 0.40
Aspartic acid/asparagine	31.09 ± 0.32
Serine	18.90 ± 0.17
Arginine	18.38 ± 0.16
Glycine	15.72 ± 0.15
Alanine	15.45 ± 0.88
Proline	11.69 ± 0.06
Threonine	9.43 ± 0.09
Cysteine	1.13 ± 0.00
EAA (% of total amino acid)	43.86
NEAA (% of total amino acid)	56.14
Umami-taste active amino acid (UAA, % of total amino acid)	35.99
Functional amino acid (FAA, % of total amino acid)	67.41
Hydrophobic amino acid (% of total amino acid)	33.85

Values are given as mean ± standard deviation from triplicate determinations. * Values reported in the parenthesis represent the percentage respect the total protein nitrogen. The umami-taste active amino acid (UAA) index was calculated by adding the individual values of glutamine, asparagine, glycine, and alanine. The functional amino acid (FAA) index was calculated by adding the individual values of leucine, threonine, methionine, arginine, glycine, alanine, proline, and cysteine. The hydrophobic amino acid index was calculated by adding the individual values of leucine, valine, phenylalanine, isoleucine, alanine, proline, and cysteine.

**Table 5 foods-10-02943-t005:** Fatty acid profile of total lipid, polar lipid fraction, and neutral lipid fraction of pig brain.

Fatty Acids (% of Total Fatty Acid)	Total Lipid	Polar Lipid Fraction	Neutral Lipid Fraction
Saturated fatty acid (SFA)			
Lauric acid (C12:0)	0.06 ± 0.09 b	0.17 ± 0.29 a	0.21 ± 0.33 a
Myristic acid (C14:0)	0.45 ± 0.31 b	0.57 ± 0.52 a	0.38 ± 0.13 b
Pentadecanoic acid (C15:0)	0.07 ± 0.02 b	0.10 ± 0.06 a	0.06 ± 0.02 b
Palmitic acid (C16:0)	14.30 ± 0.87 b	22.24 ± 10.37 a	15.49 ± 3.92 b
Heptadecanoic acid (C17:0)	0.84 ± 0.11 a	0.57 ± 0.08 b	0.52 ± 0.14 c
Stearic acid (C18:0)	17.92 ± 0.89 c	24.40 ± 14.61 a	23.12 ± 2.28 b
Arachidic acid (C20:0)	0.25 ± 0.04 a	0.23 ± 0.00 b	0.25 ± 0.06 a
Heneicosanoic acid (C21:0)	0.04 ± 0.04 c	0.08 ± 0.03 a	0.06 ± 0.03 b
Behenic acid (C22:0)	0.13 ± 0.07 c	0.21 ± 0.02 a	0.16 ± 0.11 b
Lignoceric acid (C24:0)	0.12 ± 0.07 b	0.24 ± 0.05 a	0.16 ± 0.30 b
Total SFA	34.18 ± 1.69 c	48.81 ± 2.27 a	40.43 ± 3.21 b
Monounsaturated fatty acid (MUFA)			
Palmitoleic acid (C16:1 *n*-7)	1.09 ± 0.27 a	0.91 ± 0.58 b	0.72 ± 0.20 c
Elaidic acid (C18:1 *n*-9 trans)	0.06 ± 0.04 c	0.30 ± 0.10 a	0.10 ± 0.12 b
Cis-9-octadecenoic acid (C18:1 *n*-9)	20.40 ± 1.17 c	25.28 ± 13.38 b	27.35 ± 5.38 a
Cis-11-eicosenoic acid (C20:1 *n*-11)	2.02 ± 0.43 a	1.01 ± 1.11 c	1.59 ± 0.32 b
Total MUFA	23.60 ± 1.71 c	27.50 ± 10.15 b	29.76 ± 0.16 a
Polyunsaturated fatty acid (PUFA)			
Cis-9,12-octadecadienoic acid (C18:2 *n*-6)	1.41 ± 0.27 a	0.82 ± 0.80 c	1.13 ± 0.31 b
Cis-9,12,15-octadecatrienoic acid (C18:3 *n*-3)	0.06 ± 0.03 a	nd	0.07 ± 0.10 a
Cis-6,9,12-octadecatrienoic acid (C18:3 *n*-6)	0.05 ± 0.01 a	0.07 ± 0.47 a	0.05 ± 0.07 a
Cis-11, 14-eicosadienoic acid (C20:1 *n*-6)	0.26 ± 0.01 a	0.14 ± 0.04 c	0.18 ± 0.03 b
Cis-8, 11, 14-eicosatrienoic acid (C20:3 *n*-6)	1.57 ± 0.65 a	0.88 ± 2.33 b	0.93 ± 0.48 b
Cis-5, 8, 11, 14, 17-eicosapentaenoic acid (C20:5 *n*-3, EPA)	0.06 ± 0.35 a	nd	nd
Cis-4,7,10,13,16,19-docosahexaenoic acid (C22:6 *n*-3, DHA)	20.05 ± 1.14 a	9.13 ± 10.21 c	11.78 ± 5.78 b
Total PUFA	23.46 ± 1.78 a	11.06 ± 8.41 c	14.13 ± 2.45 b
Total *n*-3 PUFA	20.16 ± 1.15 a	9.07 ± 6.84 c	11.84 ± 2.27 b
Total *n*-6 PUFA	3.29 ± 0.95 a	1.93 ± 2.78 c	2.30 ± 0.72 b
Ratio *n*-3/*n*-6	6.12 ± 0.17 a	4.70 ± 0.21 c	5.15 ± 0.10 b

nd: not detected. Values are given as mean ± standard deviation from triplicate determinations. Different letters in the same row indicate significant differences (*p* < 0.05).

## Data Availability

Not applicable.

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
