# Peer review of "Compositional Features and Nutritional Value of Pig Brain: Potential and Challenges as a Sustainable Source of Nutrients"

_foods, 2021, doi:10.3390/foods10122943_

Round 1
Reviewer 1 Report
The manuscript entitled “Compositional Features and Nutritional Value of Pig Brain Potential and Challenges as a Sustainable Source of Nutrients”, authored by Chanted and colleagues, deals with the investigation of the nutritional compositional properties of crossbred pig brains with the aim to valorize this by-products in the sustainable meat industry.
The authors performed all the analyzes necessary to correctly perform a quality control of a potential food matrix. The data obtained are very interesting, however small considerations must be made.
- Keywords should be words not contained in the title, at most present in the abstract. Their usefulness is to make easier the searching of the article using the common scientific search engines. Since several keywords are already present in the title, and/or repeated several times in the abstract, I strongly advise the authors to replace some of them and add more. As journal guidelines clearly report, a limited number of keywords can be used (maximum 10). Consequently, authors should carefully choose them.
- Material and methods section should be more descriptive, even though the methods were previously published in other articles.
- Data reported in table 1 should be expressed as g (or mg) of a specific compound per 100 grams of fresh weight. Reporting data referring first to wet weight, then to dry weight, and then to grams of lipids or sample makes very difficult understand how much compound is actually present in the food matrix.
- Moreover, after modifying data as suggested, the authors should perform a one-way ANOVA on the data reported in Table 1. The authors should remember to write the meaning of the lower case letters of the ANOVA in the legend of the table, with the respective post-hoc test.
- Color attributes and appearances of pig brain should be moved in a separate Table.
- Data reported in Table 2 should be subjected to one-way ANOVA analysis. The authors should remember to write the meaning of the lower case letters of the ANOVA in the legend of the table, with the respective post-hoc test.
- Please, modify the caption of Table 3 as follows: “Distribution of nitrogenous constituents and amino acid compositions of pig brain. Values reported in the parenthesis represent the relative percentage respect the total protein nitrogen. Functional amino acid index was calculated by adding the individual values of Leucine, Threonine, Methionine, Arginine, Glycine, Alanine, Proline, and Cysteine. Umami-taste-active amino acid index was calculated by adding the individual values of Glutamine, Asparagine, Glycine, and Alanine. Hydrophobic amino acid index was calculated by adding the individual values of Leucine, Valine, Phenylalanine, Isoleucine, Alanine, Proline, and Cysteine.
- Data reported in Table 3 should be subjected to one-way ANOVA analysis. The authors should remember to write the meaning of the lower case letters of the ANOVA in the legend of the table, with the respective post-hoc test.
- Please, report in the legend of Table 3 the unit of measure. Are data expressed as percentage?
- A solid discussion is completely missing from the article. Authors should compare their data with those obtained by other authors on other meat byproducts from the same food chain or common meat raw material.
- Conclusion section should be better discussed. It currently reports brief considerations of the data obtained and is not very useful.
Author Response
The manuscript entitled “Compositional Features and Nutritional Value of Pig Brain Potential and Challenges as a Sustainable Source of Nutrients”, authored by Chanted and colleagues, deals with the investigation of the nutritional compositional properties of crossbred pig brains with the aim to valorize this by-products in the sustainable meat industry. The authors performed all the analyzes necessary to correctly perform a quality control of a potential food matrix. The data obtained are very interesting, however small considerations must be made.
• Keywords should be words not contained in the title, at most present in the abstract. Their usefulness is to make easier the searching of the article using the common scientific search engines. Since several keywords are already present in the title, and/or repeated several times in the abstract, I strongly advise the authors to replace some of them and add more. As journal guidelines clearly report, a limited number of keywords can be used (maximum 10). Consequently, authors should carefully choose them.
Ans: Keywords were modified as suggested. Thank you very much.
• Material and methods section should be more descriptive, even though the methods were previously published in other articles.
Ans: More detailed methods were added to the material and methods section.
• Data reported in table 1 should be expressed as g (or mg) of a specific compound per 100 grams of fresh weight. Reporting data referring first to wet weight, then to dry weight, and then to grams of lipids or sample makes very difficult understand how much compound is actually present in the food matrix.
Ans: All of the data in Table 1 was given in g/100 g fresh weight.
• Moreover, after modifying data as suggested, the authors should perform a one-way ANOVA on the data reported in Table 1. The authors should remember to write the meaning of the lower case letters of the ANOVA in the legend of the table, with the respective post-hoc test.
Ans: A one-way ANOVA was employed to perform the statistical analysis, and different letters were used to indicate significant differences (p < 0.05).
• Color attributes and appearances of pig brain should be moved in a separate Table.
Ans: Pig brain color attributes and appearances were relocated to Table 2, and statistical analysis was performed.
• Data reported in Table 2 should be subjected to one-way ANOVA analysis. The authors should remember to write the meaning of the lower case letters of the ANOVA in the legend of the table, with the respective post-hoc test.
Ans: Table 2 was changed to Table 3 and the statistical analysis was performed.
• Please, modify the caption of Table 3 as follows: “Distribution of nitrogenous constituents and amino acid compositions of pig brain. Values reported in the parenthesis represent the relative percentage respect the total protein nitrogen. Functional amino acid index was calculated by adding the individual values of Leucine, Threonine, Methionine, Arginine, Glycine, Alanine, Proline, and Cysteine. Umami-taste-active amino acid index was calculated by adding the individual values of Glutamine, Asparagine, Glycine, and Alanine. Hydrophobic amino acid index was calculated by adding the individual values of Leucine, Valine, Phenylalanine, Isoleucine, Alanine, Proline, and Cysteine.
Ans: Table 3 was changed to Table 4. The caption was modified as suggested. Thank you very much.
• Data reported in Table 3 should be subjected to one-way ANOVA analysis. The authors should remember to write the meaning of the lower case letters of the ANOVA in the legend of the table, with the respective post-hoc test.
Ans: Table 3 was changed to Table 4 and the statistical analysis was performed.
• Please, report in the legend of Table 3 the unit of measure. Are data expressed as percentage?
Ans: Table 3 (which is now Table 4) has a well explained legend. The nitrogenous constituents are expressed in milligrams N per gram of sample. The protein nitrogen values in parenthesis represent the relative percentage of total protein nitrogen. Amino acid compositions are expressed in milligrams per gram of protein. The EAA, NEAA, UAA, FAA, and hydrophobic amino acid indices were calculated using the percentage of total amino acid.
• A solid discussion is completely missing from the article. Authors should compare their data with those obtained by other authors on other meat byproducts from the same food chain or common meat raw material.
Ans: Due to the limited information regarding the composition of animal brains, we tried to compare the results with the compositions of other animal-derived by-products, meat raw material, and other foodstuffs. The discussion was revised accordingly. Thank you very much.
• Conclusion section should be better discussed. It currently reports brief considerations of the data obtained and is not very useful.
Ans: The conclusion was revised as suggested.

Reviewer 2 Report
Review ID: foods-1472295
Ten brains of hybrid pigs aged 4 months were used for the study. Although the scope of the study is quite wide, the paper is in my opinion only informative. The authors did not sufficiently emphasise the novelty of their research. The commercial and nutritional value of brains is well known and they are fully utilised in meat industry. It is a pity that the authors did not present comparative studies, analysing the influence of e.g. breed, feeding, system of keeping pigs, remaining with statistics, and this to a basic level.
The reviewed article is generally methodologically correct, although it lacks the names and manufacturers of the analytical apparatus used in sections 2.3, 2.4, 2.5, 2.6, 2.7. Additional information should also be completed in section 2.4 regarding the modification of the Beyer and Jensen [1987] method, the colour space (Hunter, CIE) in section 2.5, or the determination of the fatty acid composition (fatty acids or esters) in section 2.9. No control group available.
As presented, the paper does little to advance current knowledge of the nutritional value of edible by-products. In short, the work is only causative. I suggest that the authors complement the article with a tabular comparison of the results obtained with those obtained by other authors for the brains of other species of slaughter animals, as an attempt to highlight the differences or greater nutritional value of pig brains.
Author Response
Ten brains of hybrid pigs aged 4 months were used for the study. Although the scope of the study is quite wide, the paper is in my opinion only informative. The authors did not sufficiently emphasise the novelty of their research. The commercial and nutritional value of brains is well known and they are fully utilised in meat industry. It is a pity that the authors did not present comparative studies, analysing the influence of e.g. breed, feeding, system of keeping pigs, remaining with statistics, and this to a basic level.
Ans: Of course, the commercial and nutritional value of brains is well known, and they are widely used in the meat industry, but there is no published research article on the composition of pig brains. As a result, we attempted to determine the compositional characteristics of pig brain as well as its nutritional value. The impacts of breed, nutrition, and pig-keeping systems on pig brain composition are all things that will be investigated further in the future. Thank you so much for invaluable suggestion.
The reviewed article is generally methodologically correct, although it lacks the names and manufacturers of the analytical apparatus used in sections 2.3, 2.4, 2.5, 2.6, 2.7. Additional information should also be completed in section 2.4 regarding the modification of the Beyer and Jensen [1987] method, the colour space (Hunter, CIE) in section 2.5, or the determination of the fatty acid composition (fatty acids or esters) in section 2.9. No control group available.
Ans: More detailed methods were added to the material and methods section.
As presented, the paper does little to advance current knowledge of the nutritional value of edible by-products. In short, the work is only causative. I suggest that the authors complement the article with a tabular comparison of the results obtained with those obtained by other authors for the brains of other species of slaughter animals, as an attempt to highlight the differences or greater nutritional value of pig brains.
Ans: Due to the limited information regarding the composition of animal brains, we tried to compare the results with the compositions of other animal-derived by-products, meat raw material, and other foodstuffs. The discussion was revised accordingly. Thank you very much.

Round 2
Reviewer 2 Report
Authors have improved the manuscript very well, taking into account almost all my comments and suggestions as well as giving explanations in disputable cases. Overall, authors have done a good job, thus I have no further points to make.
Author Response
Thank you very much.